# Metaheuristic for Optimal Dynamic K-Coloring Application on Band Sharing for Automotive Radars

**DOI:** 10.3390/s23125765

**Published:** 2023-06-20

**Authors:** Sylvain Roudiere, Vincent Martinez, Pierre Maréchal, Daniel Delahaye

**Affiliations:** 1Artificial and Natural Intelligence Toulouse Institute, Université Fédérale Toulouse Midi-Pyrénées, 31000 Toulouse, France; 2NXP Semiconductors, 31100 Toulouse, France; vincent.martinez@nxp.com; 3Mathematical Institute of Toulouse, 31400 Toulouse, France; pierre.marechal@math.univ-toulouse.fr; 4École Nationale de l’Aviation Civile, 31400 Toulouse, France; daniel@recherche.enac.fr

**Keywords:** automotive, channel access policy, cooperation, genetic algorithm, interference mitigation, metaheuristic, optimization, radar, simulated annealing, V2X

## Abstract

The number of vehicles equipped with radars on the road has been increasing for years and is expected to reach 50% of cars by 2030. This rapid rise in radars will likely increase the risk of harmful interference, especially since radar specifications from standardization bodies (e.g., ETSI) provide requirements in terms of maximum transmit power but do no mandate specific radar waveform parameters nor channel access scheme policies. Techniques for interference mitigation are thus becoming very important to ensure the long-term correct operation of radars and upper-layer ADAS systems that depend on them in this complex environment. In our previous work, we have shown that organizing the radar band into time-frequency resources that do not interfere with each other vastly reduces the amount of interference by facilitating band sharing. In this paper, a metaheuristic is presented to find the optimal resource sharing between radars, knowing their relative positions and thereby the line-of-sight and non-line-of-sight interference risks during a realistic scenario. The metaheuristic aims at optimally minimizing interference while minimizing the number of resource changes that radars have to make. It is a centralized approach where everything about the system is known (e.g., the past and future positions of the vehicles). This and the high computational load induce that this algorithm is not meant to be used in real-time. However, the metaheuristic approach can be extremely useful for finding near optimal solutions in simulations, allowing for the extraction of efficient patterns, or as data generation for machine learning.

## 1. Introduction

The use of advanced driver assistance systems (ADAS) in the automotive domain has been expanding significantly over the past decade. These systems are expected to be present in 700 millions cars by 2030 [1,2]. This means that approximately 50% of cars will be equipped with radars. They will enable more secure road traffic by anticipating potential collisions and ensure safer driving. However, a rapid increase in radar on the road without regulation on how to share the available radar band will likely lead to harmful interference being generated [2], whose effects can vary from reducing the range of the radar, blinding it, or generating false targets [3]. Multiple projects have been started to investigate the automotive radar interference mechanisms, such as the European funded project MOSARIM [2] and the publicly funded German project IMIKO [4].

Automotive radars’ self-interference is a generic problem that applies to all of the allocated radar bands. The 24 GHz band has been phased out since 2022 [5], and automotive radar applications have been moved to the 76–81 GHz band. Regional standardization bodies such as ETSI in Europe provide requirements for the maximum peak and mean transmit powers [6] (the different values for the 76–77 GHz and 77–81 GHz bands [7,8]). Additionally, new bands around 140 GHz are being investigated [9] and arguably bring fundamentally similar challenges. However, these automotive radar standards do not mandate specific channel access scheme policies nor specific radar waveforms, thereby leading to potential interference due to the absence of coordination between the actors.

Automotive radars are used for a wide range of driving assistance applications. Long-range radars (LRR) are commonly used for adaptive cruise control (ACC) or automatic emergency braking (AEB), whereas medium and short range radars (MRR/SRR) find application in blind spot detection (BSD) or lane change assist (LCA), for example. Depending on the application, radars will have different characteristics in terms of range and velocity measurements and field-of-view.

Moreover, extensive research is being conducted on new types of hardware and antenna systems for automotive radars. For instance, there is extensive research on the topic of imaging radars [10,11] or SAR [12,13,14], as well as other techniques such has MIMO radars.

This paper focuses on LRR-frequency-modulated continuous wave (FMCW) radars as they are the most common today, but the metaheuristic proposed can be applied to any type of radar, including non-vehicular radars such as road-side units.

Conversely, our previous work on the subject [15], and ref. [16] focused on avoiding interference using a communication channel such as V2X (vehicle to everything). V2X usually defines an ad hoc network in the 5.9 GHz ITS (intelligent transport services) band, and regional implementations (e.g., ETSI ITS-G5 in Europe) can be based either on the IEEE 802.11p/bd Wi-Fi standard, or on the 3GPP standards such as 4G LTE-V2X and 5G NR V2X [17]. It might also be possible to use 6G joint data communication and sensing (JCAS) in the future, a topic being vastly investigated [18], with some potential for automotive ADAS applications. In short, we explored the possibility to avoid interference from happening, as opposed to the class of signal processing techniques that aim at repairing the signal once interference has already happened.

In this paper, a graph coloring approach is explored to minimize the amount of interference. Indeed, when considering the use of orthogonal waveform by the radars (two radars using orthogonal waveforms do not interfere with each other), avoiding all interference can be translated into a graph coloring problem where the nodes are the radars, their colors are the waveform used, and the edges are the line-of-sight between them. Considering that road traffic is not stationary and that the number of available orthogonal waveforms is finite and known, the problem is similar to a dynamic K-coloring problem. Finding a K-coloring for a given snapshot of the road traffic can be quite easy depending on its complexity and the number of colors/waveforms available. That is why, in addition to avoiding interference, the metaheuristic presented in this paper also aims at minimizing the number of color changes made by each radar as it increases the stability and predictability of the system.

The results obtained are close to the optimal radar band sharing as they avoid all interference while being extremely stable. The metaheuristic method cannot be translated directly into a mitigation strategy as it requires the knowledge (present and future) of all radars, their line-of-sight, and the environment, and it also requires a lot of computation power. However, the results are close to the optimal radar band sharing as they avoid all interference while being extremely stable. These results can then be used to extract patterns or as data input for advanced AI techniques. As the metaheuristic supposes a perfect knowledge of the environment, future techniques using it for training would benefit a lot from technologies such as collective perception and sensor fusion to have a close enough understanding of the environment.

The paper is organized as followed. Section 2 will present the current state-of-the-art on graph coloring algorithms. Section 3 details the simulation environment built to extract the graph on which the metaheuristic is applied. Section 4 describes how each of the graph’s components are built. The metaheuristic is shown in Section 5 with its results in Section 6, followed by a conclusion in Section 7.

## 2. State of the Art

### 2.1. Automotive Radar Interference Mitigation

Current automotive radar interference mitigation strategies rely for the most part on signal processing to repair the signal that has been interfered with. These methods rely on the system’s capabilities to detect interfered samples and reconstruct the interference-free signal, or at least reduce the impact of the interference on the overall radar performances. These strategies can work very well when dealing with few interferers, but as automotive radar usage is predicted to grow in the coming years [1], these techniques might not be enough. Multiple projects such as MOSARIM [2] and IMIKO [4] have identified categories of interference mitigation strategies with their respective estimated mitigation power. These techniques include but are not limited to changing chirps transmit frequency in presence of interference, randomizing chirps length, repairing interfered signals, or using specific polarization depending on the radar location on the car.

Some papers are studying the possibility of using a communication channel to define a channel access policy, such as RadChat [19], which proposes a new radar with communication capabilities included in its design to share predefined radar band resources. RadarMAC [20], on the other hand, takes a more centralized approach where vehicles are connected to a base station that orchestrates the sharing of the available resources depending on the radars positions. These techniques yield reductions in the amount of interference but do not focus on finding the theoretical optimal sharing of the radar band.

### 2.2. The Vertex Coloring Problem

The vertex coloring problem (VCP) has received a lot of attention for real world applications in many different fields of engineering such as scheduling [21], register allocation [22], communication networks [23], or frequency assignment [24], just to name a few of them. The VCP is known to be NP-hard [25], meaning that only small instances (<100 vertices) can be solved with an exact algorithm in a reasonable computation time. However, real world applications commonly deal with a much higher number of vertices, as is the case for automotive radars.

There are other approaches to the VCP when dealing with instances too large to be solved with exact methods. Greedy algorithms such as SEQ or RLF [21] give a correct coloring in a very short amount of time, but they are usually far from the best solution as they do not try to find the optimum but only a good quality solution. On the other hand, heuristics and metaheuristics aim at finding a close to optimal solution by using local search methods to explore the solution space. Some well-known ones are DSATUR [26], a heuristic that sequentially colors the vertices based on the degree of saturation, and TABUCOL [27], a metaheuristic that uses tabu-search for its coloring. Some metaheuristics use simulated annealing [28] or genetic algorithm [29] to iteratively find better solutions.

Coloring a dynamic graph whose edges and vertices can be inserted or removed from one timestep to another is a different problem. One way to approach the problem is to use the previously mentioned algorithm at each timestep, recomputing a new coloring from the ground up every time a change is made to the graph. This method is time-consuming as heuristics and metaheuristics usually need some computation time, but it will yield good coloring at each timestep. This also will not take into account the similarities between two consecutive timestep.

Moreover, other than the number of colors used, dynamic graph coloring introduces a new metric: the number of color changes between two timesteps. By minimizing the number of color changes, the solution found at a given timestep will not be very different from the solution at the next one. In the case of automotive radars, this means that radars (vertices) do not have to change their waveforms (colors) very often. This increases the stability of the radar band sharing, as well as increasing its predictability and its robustness to the loss of the communication signal. Current algorithms are designed around this trade-off between the number of colors and the number of re-colorations [30]. For example, ref. [31] introduces two algorithms. The first one, “small-buckets algorithm”, maintains, for any integer d>0, a proper $O(CdN^{1/d})$-coloring while recoloring, at most, O(d) vertices, with *C* and *N* representing the maximum chromatic number and the maximum number of vertices respectively. The second algorithm, “big-buckets algorithm”, reverses the trade-off by maintaining an O(Cd)-coloring with $O(dN^{1/d})$ re-colorings.

In our case of automotive radars, the available radar band being fixed means that the number of available colors is also fixed. There is no need for a trade-off; instead, the goal is to keep a K-coloring while minimizing the number of re-colorings. In order to find this optimal coloration, a new algorithm must be designed.

## 3. Simulation Environment

Our previous works [15,32] focused on V2X-based interference mitigation strategies with a Python simulator, which was built to generate realistic traffic scenarios including radars. This framework uses the recognized and widely used SUMO software (SUMO is an open-source continuous multi-modal and highly portable simulation package made with Eclipse. It is designed to handle large road networks and is already used by some V2X studies to provide realistic road traffic scenarios. The software is available at https://www.eclipse.org/sumo/ (accessed on 1 June 2023)) to generate the road traffic (the configuration used in SUMO is available online at https://github.com/Ramboun/dynamic_k_coloring_data (accessed on 1 June 2023)). SUMO allows for the generation of custom scenarios, but a real traffic scenario could be used via existing real-world databases, such as the US Highway 101 dataset [33], or new data acquired with connected vehicles [34]. The scenario used in this paper is the same as in our previous work and consists of a 2 km highway with 6 lanes. Two clusters of cars are generated at each end totaling 151 vehicles going from one end to the other of the highway.

Once the position and orientation of the cars are known at any given timestep, a frequency-modulated continuous wave (FMCW) long-range radar (LRR) is added to the front of each car. Other types of radar such as MRR or SRR would follow similar procedures, although their field-of-view and range would be different, but they would be handled by the metaheuristic. LRRs are being used as they are the most common nowadays. Typical LRR parameterization has been assumed, with 20° field-of-view and a maximum range of 300 m. The available radar band is organized into multiple time-frequency zones, sometimes also referred to as resources, which can be chosen by radars to emit in. Each time-frequency zone is defined with to respect the desired range and speed resolution that an LRR requires. With each zone being separated, interference can only occur between radars using the same zone. It must be clear that the present paper does not promote a specific way to organize the radar band, and the metaheuristic approach only looks at it in terms of orthogonal resources. However, Figure 1 and Figure 2 are provided to exemplify possible organizations of the radar band. Figure 1 is an example of a very constrained band split used in our previous work [16], but a simple approach to radar band organization could be a grid-like split as illustrated in Figure 2. The number of available resources depends on the shape of these time-frequency zones.

Finally, two radars can interfere only if they “see” each other. They either need to be in direct line-of-sight (LOS) or in non-line-of-sight indirect connection (sometimes also referred to as indirect line-of-sight (ILOS)) resulting from the signal bouncing around. LOS is modeled by simply checking if two radars are oriented towards each other, taking into account their field-of-view. ILOS situations are more complicated to model as they depend on the reflecting surface characteristics. When a target is within the field-of-view of a radar, a ray is cast from the radar to the target, and the point where the ray meets the target will be the source of an echo at 180° toward the normal of the surface hit. These computations of line-of-sight are illustrated in Figure 3. Figure 4 shows a snapshot of the simulation, with the blue rectangle representing cars and the green (LOS) and red (ILOS) lines representing the line-of-sights between their radars.

## 4. Mathematical Modeling

### 4.1. Dynamic Graph K-Coloring

Interference mitigation between radars can be achieved by avoiding the detrimental situations where two radars that are in LOS (or ILOS) are using this same time-frequency resource. This can be translated into a dynamic graph K-coloring problem. Let K be a set of colors representing the different available time-frequency zones. Let R be a set of vertices corresponding to the different radars of our simulation, and let T be the set of timesteps. We can define the temporal adjacency matrix A(R,R,T) where $a_{i,j,t}=1$ only if the radars *i* and *j* are in line-of-sight at the timestep *t* of the simulation. The objective is then to find a valid K-coloring of the graph at every timestep while minimizing the number of re-colorations needed. By defining $x_{r,k,t}$ to be equal to 1 if and only if the radar *r* uses the color *k* at timestep *t*, and 0 otherwise, we end up with the following problem formulation: (1)$$\begin{array}{cccc} \; & \;\min & \; & \sum_{t\in T^{\prime }}\sum_{r\in R}\sum_{k\in K}\left|x_{r,k,t+1}-x_{r,k,t}\right| \end{array}$$(2)$$\begin{array}{ccccc} \; & \;s.t. & \; & \sum_{k\in K}x_{r,k,t}=1 & \forall r\in R,\forall t\in T \end{array}$$(3)$$\begin{array}{cccc} \; & \; & & \sum_{r_{i}\in R}\sum_{r_{j}\in R}\sum_{k\in K}\sum_{t\in T}A_{r_{i},r_{j},t}*x_{r_{i},k,t}*x_{r_{j},k,t}=0 \end{array}$$(4)$$\begin{array}{ccccc} \; & \; & & x_{r,k,t}\in \left\{0,1\right\} & \forall r\in R,\forall k\in K,\forall t\in T \end{array}$$(5)$$\begin{array}{ccccc} \; & \; & & A_{r_{i},r_{j},t}\in \left\{0,1\right\} & \forall r_{i}\in R,\forall r_{j}\in R,\forall t\in T \end{array}$$
where $T^{\prime }=\left\{0,\ldots ,t_{T-1}\right\}$.

The objective function (Equation 1) ensures a minimum amount of re-coloration between any two timesteps *t* and t+1. Constraint (Equation 1) forces the radars to use only one color at a time. Constraint (Equation 1) avoids any interference by forcing two radars to have a different colors if they are in line-of-sight. To satisfy constraint (Equation 1), the number of available colors needs to be equal to or larger than the graph chromatic number.

### 4.2. Optimization Mathematical Model

Metaheuristics are algorithms designed to find good solutions to optimization problems where finding the optimum is near impossible due to the size of their solution space. By exploring the solution space in a controlled manner, metaheuristics aim at gradually obtaining the optimal solution.

The metaheuristic presented in this paper views radars as agents that, one by one, pick the best possible color choices (with their limited knowledge of other radars’ choices) at every timestep. The colors chosen by a given radar will impact the colors picked by radars going after it as they will try to avoid conflicting with it. This process can be translated into the following framework:

Consider a set of colors K, a set of radar identifiers R, a set of timestamps $T=\{t_{1},\ldots ,t_{end}\}$, and the temporal adjacency matrix *A* describing the line-of-sight between radars through time. Additionally, consider the oriented graph $G=(V,E_{changes},E_{conflicts})$, where each node represents a possible state for a radar during the simulation. V=S∪E∪C with:$S=\{S_{r}|r\in R\}$ the individual starting state of each radar *r*. This state is not associated with any color. It corresponds to the start of the simulation where the first colors used by each radar are not determined yet.$E=\{E_{r}|r\in R\}$ the individual ending state of each radar *r*. Like the starting state, it is not associated with any color and represents the end of the simulation.$I=\{I_{r,t,k}|r\in R,t\in T,k\in K\}$ the set of intermediary possible states for each radar *r* other than starting and ending states. At timestep *t*, a radar *r* using color *k* will be in the state represented by the node $I_{r,t,k}$.

The set of edges $E_{changes}$ represents the possible changes of state for a radar during the simulation:(6)$$\begin{array}{cc} E_{changes} & =\{\left(I_{r,t,k_{1}},I_{r,t+1,k_{2}}\right)\;|\;r\in R,t\in T-\left\{t_{end}\right\},k\in K,\forall \left(k_{1},k_{2}\right)\in K^{2}\}\\ \; & \;\cup \{\left(S_{r},I_{r,t_{1},k}\right)\;|\;r\in R,k\in K\}\\ \; & \;\cup \{\left(I_{r,t_{end},k},E_{r}\right)\;|\;r\in R,k\in K\} \end{array}$$

Our goal is to minimize the number of changes necessary to keep our K-coloring. The weight of an edge is determined as follows:(7)$$Cost\left(\left(v_{1},v_{2}\right)\in E_{changes}\right)=\left\{\begin{array}{cc} \;1 & \;ifv_{1}=I_{r,t,k_{1}},\\ \; & \;v_{2}=I_{r,t+1,k_{2}},k_{1}\neq k_{2}\\ \;0 & \;else. \end{array}\right.$$

Let $P_{r}=\left(S_{r},I_{r,t_{1},k_{1}},I_{r,t_{2},k_{2}},\ldots ,I_{r,t_{end},k_{end}},E_{r}\right)$ be the path taken by radar *r* across the graph, describing, at each time, which color it took, and let $L_{r}$ be its cost. The cost of the path will be impacted by the edges taken but also by the nodes that are in conflict.

We can define the set of conflicting nodes of a radar $r_{1}$ at time *t* with color *k* by:(8)$$\begin{array}{c} V_{conf}\left(r_{1},t,k\right)=\left\{I_{r_{2},t,k}\;|\;A_{r_{1},r_{2},t}=1\right\} \end{array}$$

Let β be the cost of a conflict; we have the following cost for each node:(9)$$Cost\left(I_{r_{1},t,k}\right)=\sum_{I_{r_{2},t,k}\in V_{conf}\left(r_{1},t,k\right)}\beta *p\left(I_{r_{2},t,k}\right)$$
where:(10)$$p\left(I_{r,t,k}\right)=\left\{\begin{array}{cc} \;1 & \;\mathrm{if}I_{r,t,k}\in P_{r}\\ \;0 & \;\mathrm{else}. \end{array}\right.$$

Minimizing the number of color changes by the radars while avoiding conflicts at all cost corresponds to the following objective:(11)$$\min\sum_{r\in R}L_{r}$$
with $L_{r}$ being the cumulative cost of all of the edges and nodes in the taken path $P_{r}$ with β>|R||T|. Indeed, with β>|R||T|, a conflict cost more than a color change by every radar at every timestep. A solution with a lower number of conflicts will always have a lower total cost compared to a solution with a higher number of conflicts, and this will be true whatever the number of re-colorations needed. Having β<=|R||T| is a way to allow conflicts in the case where avoiding them would require too many re-colorations.

Figure 5 is an example of what the resulting coloring graph looks like.

## 5. Metaheuristic

### 5.1. Principle

Using the previously defined metaheuristic graph, finding a solution to our dynamic K-coloring problem for one of the radar $r_{i}$ is as simple as “dropping a marble” on its sub-graph and letting it fall from $S_{i}$ to $E_{i}$, following the shortest path through the sub-graph. It will trigger an increase in cost for nodes that are in conflict with the ones that are part of its path. Repeat this operation for every sub-graph to obtain a first solution to our problem.

The marble dropping mechanic is implemented by solving a shortest path problem within each sub-graph, one by one. Because of the sub-graphs structure, the Bellman–Ford algorithm is particularly efficient to compute the shortest path. Indeed, because of the absence of cycle or loop (edges only go from a timestep *t* to t+1), this is the best case scenario for the Bellman–Ford algorithm, with a time complexity in O(|E|), with |E| representing the number of edges ran through.

The solution found at the end depends on the order in which the different sub-graphs’ shortest path are computed. Once the shortest path is found, the path is checked again to update the cost of conflicting neighbors nodes within the other sub-graphs. The presence of conflicting neighbors depends on the temporal adjacency matrix *A*. Increasing the cost of certain nodes in other sub-graphs will impact their shortest path computation, hence the importance of the computation order. The increase in cost is known a posteriori. It is possible for a sub-graph to generate cost on nodes that are already being used within another sub-graph’s path. This is why the length of each path (not the path itself) needs to be computed again at the end to take into account these changes.

Changing the order of “marble drop” will lend many different solutions, but some solutions cannot be reached. Indeed, the solution will always have a radar that never changes its color. The first sub-graph to compute its path will not have any cost increase on its nodes. It will then choose one color and keep it until the end as the cost will remain 0. To increase the number of possible solutions found by the metaheuristic and include a solution with at least one color change for every radar, it is necessary to introduce *Gates*.

*Gates* can be placed at any timestep and act as a stopping point for the marbles. Each gate has its own order permutation. Instead of computing the shortest path for the entire sub-graph in one shoot, the shortest path between a gate and the next one is computed before starting again from the new gate but with a new order permutation. An example of a gate effect on the solution is shown in Figure 6.

In a scenario with *R* radars, *T* timesteps, and *K* colors, the total number of edges in our coloring graph is $R*(\left(T-1\right)*K^{2}+2K)$. The time complexity for computing a solution is then in $O(RK^{2}T)$.

To optimize the computation time, it is recommended to smash the input graph in some places. For example, if during multiple timesteps there are no new edges, then the K-coloring found for the first of these timestep will be valid for every one of them. Thus, it is interesting to smash these timesteps into one to shorten the computation time. However, it is necessary to add a multiplying factor to the cost of this smashed timestep as a conflict on it represents a conflict for multiple timesteps.

### 5.2. Sliding Window Adaptation

When computing the shortest path across an entire sub-graph, the algorithm simulates the fact that a radar knows exactly what the future line-of-sights with other radars will be, as well as the colors they will be using (depending on the permutation order). It is useful to find the optimal color sharing amongst radars, but limiting this knowledge of the future can yield results that would be easier to replicate in real life scenarios.

To achieve this, the time window adaptation of the coloring graph limits the number of timesteps to be considered when solving the shortest paths. To determine colors used at timestep *t*, a new graph is extracted from the main one, containing all of the timesteps from *t* to t+W, with *W* the time window size. The previous timestep t−1 is also included but will not be modified; it is used to keep track of the color of each radar before reaching timestep *t*.

The metaheuristic can then be performed on this new smaller graph to find the best color sharing. Once it has been performed, the color chosen at timestep *t* is locked, and the process repeats for the timesteps t+1 to t+1+W and so on until every timestep has been locked.

This variant does not require the use of *gates* because the permutation order can already be different from one timestep to another. Indeed, performing the metaheuristic of the time window [t,t+W] with a certain permutation order only locks the color chosen at timestep *t*. When performing the metaheuristic on the next time window [t+1,t+1+W], the permutation can be different for timestep t+1.

The addition of the window increases the complexity compared to the non-windowed one. Solving the small graph has a complexity of $O(RK^{2}W)$, but it must be carried out for every timestep for a final time complexity of $O(RK^{2}WT)$.

The two variants are functionally different as they do not explore the same solution space. The first one explores the solution to the entire scenario, whereas the windowed variant explores solutions for smaller problems but ends up combining them to generate a single final solution to the entire problem.

### 5.3. Simulated Annealing

The first implementation of the metaheuristic uses simulated annealing [35] to explore the solution space. Simulated annealing is inspired from annealing in metallurgy, where a material is heated up and cooled down in a controlled manner to alter its physical properties.

Simulated annealing aims at finding a quasi optimum solution by going from one solution to another, depending on their respective performances but also on a parameter called temperature. A higher temperature yields a higher chance of accepting a worse solution. By starting with a high temperature and slowly decreasing it throughout the whole process, simulated annealing avoids local optima and converges toward a solution close to the global optimum.

The different aspects of this papers’ implementation of simulated annealing are as follows:**Solution**: a solution for our simulated annealing implementation is a list of size *T* containing or not a gate for each timestep as illustrated in Figure 7. Only the first timestep always has a gate that cannot be removed. As the Bellman–Ford algorithm is deterministic, the same list of gates will yield the same result (if the edge and conflict costs are the same).**Evaluation**: for each gate, the Bellman–Ford algorithm is applied to each sub-graph in the order of the gate to find the shortest path to the next gate (or to the end). The costs of conflicting nodes are updated once the shortest path is found for a sub-graph. The cost of a solution is the sum of the cost of all of the sub-graphs. As the cost of a conflict is changing, it is important to keep track of the number of conflicts of a solution so to not have to re-evaluate it entirely when the cost of a conflict is lowered. It allows one to fairly compare two solutions without being biased by the lower cost of a conflict for one of them.**Neighborhood operator**: to select a neighboring solution, multiple modifications can be carried out as illustrated in Figure 8. First, add or remove a random gate. This will remix completely the sub-graph running order. By extracting the cost of each timestep, it is possible to weight the random generation of the gate to focus more on high cost timesteps.Second, switch the position of two “gates”. This is achieved by selecting a gate at random (weighted by the timestep costs following it) and moving it to another timestep. If a gate is already present in this timestep, then both gates switch.Finally, change the order of a gate. This is achieved by selecting a sub-graph at random (weighted by its cost) and moving it earlier in the order. The earlier a sub-graph is in the permutation order, the fewer constraints it will have for its shortest-path. Since the node’s cost within a sub-graph depends on the path taken by other sub-graph before it, going early means that fewer sub-graphs have already chosen a path and thus have affected the nodes’ cost.**Temperature and conflict cost**: temperature is decreased by factor α every $n_{iter}$ iterations until the temperature reaches $T_{end}$ (usually $0.001*T_{init}$). At the same time temperature is lowered, the cost of a conflict increases by a factor $\alpha_{c}$. This factor is designed for conflict cost to reach $T*R*cost_{edge}$ near the end of the simulated annealing as T∗R is the maximum number of color changes possible (one color change by each radar at each timestep). When the cost of a conflict is equal to $T*R*cost_{edge}$, a solution having fewer conflicts than another is ensured to achieve a lower cost whatever the number of color changes. Having a low cost for a conflict at the beginning of the simulated annealing favors the exploration of the solution space by avoiding local optima with a low number of conflicts and a high number of color changes. The conflict cost is then increased as the goal is to find a solution with the minimum number of color changes while still avoiding every conflict.

The pseudo-code of this version is available in Appendix A. The main limitation with this implementation is that it cannot be parallelized. Indeed, since the shortest path of a sub-graph depends on the shortest path of the previous sub-graph, it is not possible to apply the Bellman–Ford algorithm in parallel to all sub-graphs to speed up the solution evaluation. This can be an issue when performing the windowed simulated annealing as its complexity can be orders of magnitude higher than the non-windowed one, depending on the window size. This is why the windowed simulated annealing has not been implemented in this paper. With a computation time of 1.5 weeks for the non-windowed simulated annealing, running it with a sliding window of size 10 would have required around 3.5 months of computation. Instead, another algorithm using a genetic approach is used for the sliding window coloring graph as it can be parallelized.

### 5.4. Genetic Algorithm

This second implementation of the algorithm uses a genetic approach to explore the solution space. First introduced in [36], genetic algorithms are methods inspired by natural selection to make a population of solutions evolves towards the global optimum.

The genetic algorithm achieves this by selecting the best candidates within the population, mixing them together to generate “children” solutions, adding a random mutation to some of them, and starting over with the new population made of the best candidates and the children they generated.

The different aspects of this papers’ implementation of the genetic algorithm are as follows:**Solution**: A solution for our genetic algorithm is a permutation order for the first gate. Since this genetic algorithm is to be used on the sliding window coloring graph, it does not require the use of additional gates for the reason mentioned in Section 5.2.**Evaluation**: The evaluation of a solution is the same as for the simulated annealing version. Following the order of the starting gate, each small sub-graph computes its shortest path using the Bellman–Ford algorithm. The cost of the solution is the sum of the conflict cost and color change cost of each small sub-graph.**Selection**: The selection is achieved by a deterministic tournament of size 2 without replacement. Each solution is paired randomly, and the one with the lower cost is the winner.**Crossover**: Crossover is carried out by using the position-based crossover operator (POS). It functions by randomly selecting a subset of the permutation order of the parent P1 and copying it into the child order. Then, the blanks in the child permutation order are filled in the order of the permutation of the second parent P2. This is described in Figure 9.This is then repeated again, but by copying from P2 first and filling the blank following the order of P1 to generate a second child solution.**Mutation**: Each children solution has a probability $p_{mutation}$ to mutate to favor the exploration of the solution space. This mutation is implemented with the reverse sequence mutation (RSM) as it is a well performing mutation operator on the travelling salesman problem [37] (and the TSP uses the same solution formulation of a permutation order). This mutation takes a random section of the order and reverses it as illustrated in Figure 10.**Conflict cost**: The conflict cost is handled in the same way as for the the simulated annealing version. It starts low to avoid local optima that have a low number of conflicts, and then it gradually increases up to $W*R*cost_{edge}$ in order to find a solution that does not contain any conflict by the end of the metaheuristic. Compared to the non-windowed variant, the sliding window adaptation solves smaller graphs of a length *W*; thus, the maximum cost is $W*R*cost_{edge}$ instead of $T*R*cost_{edge}$.

The pseudo-code of this version is available in Appendix B. Unlike the simulated annealing version, this genetic algorithm can be parallelized by simply evaluating the different solutions in parallel due to the fact they are independent of one another. This greatly speeds up the computation and is particularly useful for the windowed version of the metaheuristic.

However, depending on the implementation, it might require for the different process to access the same graph, which would lead to an overhead when accessing the memory. This overhead might not be negligible as the Bellman–Ford algorithm is already fast due to our graph structure. To avoid this, each process is given its own copy of the graph, eliminating this overhead but multiplying the memory usage. This is why the genetic algorithm has not been applied to the non-windowed version as not enough memory was available to store the entire graph multiple times.

## 6. Results

### 6.1. Color Changes Lower Bound

A lower bound for the number of changes necessary to correctly K-color the entire temporal graph can be determined with its smashed graph. The smashed graph of our temporal graph is the graph containing all of its nodes and all of its edges as long as they are present in at least one of the timesteps. The resulting adjacency matrix is $A^{\prime }\left(R,R\right)$ where:(12)$$a_{i,j}^{\prime }=\left\{\begin{array}{cc} \;1 & \;\mathrm{if}\exists \;t\in T/a_{i,j,t}=1\\ \;0 & \;\mathrm{else}. \end{array}\right.$$

If there is an edge between radars *i* and *j* in this smashed graph, it means that there is at least one timestep where they are adjacent. Finding the correct coloring of this smashed graph results in zero color changes needed in the temporal version. Let the chromatic number of this smashed graph be X; this means that with K≥X, we can find a starting configuration that results in zero color changes in the temporal graph. However, when K<X, the smashed graph cannot be colored properly. If two adjacent vertices *v* and $v^{\prime }$ in the smashed graph have the same color, they will be in conflict at least once, and to avoid this conflict, at least one color change is necessary.

If vertex *v* will change its color and is able to avoid every potential conflict, it can be removed from the smashed graph. The remaining smashed graph’s chromatic number is now at least $X^{\prime }=X-1$. If $K<X^{\prime }$, the previous operation is repeated until *K* is equal to the lower bound of the smashed graph chromatic number.

By supposing that only one color change is necessary at every step, and that removing the associated vertex is enough to lower the chromatic number by one, the lower bound for the number of color changes is X−K. As finding the chromatic number of a graph is itself an NP-hard problem, X can be replaced by its lower bound. The size of the largest clique can be used as a lower bound for X. As cliques are fully connected, a clique of a size *S* requires *S* colors to be colored properly. If our smashed graph includes a clique of size *S*, then X is greater than or equal to *S*.

In the case of the smashed graph extracted from the simulation environment described in Section 3, the largest clique has a size of 39, resulting in a lower bound for the number of color changes of 39−K with *K* available colors.

### 6.2. Metaheuristic Results

The metaheuristic has been applied to the graph extracted from the simulation environment described in Section 3 (the temporal adjacency matrix resulting from the simulation environment is available online at https://github.com/Ramboun/dynamic_k_coloring_data, accessed on 1 June 2023). Simulated annealing has been applied with the parameters presented in Table 1.

$\alpha_{c}$ is computed, so $cost_{conf}$ reaches $cost_{conf}^{end}$ by the end of the cool-down phase.

With these parameters, a total of a million solutions are tested, yielding the results presented in Table 2 and Figure 11 with 16≥K≥36. Overall, 16 is the lowest value tested as there is a timestep with a max clique of 16, making it impossible to avoid all conflicts with less than 16 colors. The results are compared with a method that changes colors right before a conflict happens and chooses the color that maximizes the amount of time before the next conflict. If every color is already taken, it picks a random one.

The number of changes of the best solution found increases exponentially when lowering the amount of available color *K*, which can be observed in Figure 11 as the results tend to follow a linear line in the log-scale plot. As the lower bound for the number of colors needed to avoid all conflicts is 16, the number of color changes associated with the values of K close to this value increase, but the number of conflicts is non zero.

With the windowed variant and the genetic algorithm approach, the results are presented in Table 3. These results have been achieved with the parameters presented in Table 4.

As expected, the number of changes found with the windowed variant is much higher than the non-windowed one. Because of the limited knowledge of the future, it is impossible for a long term strategy or pattern to emerge. The choice of color at timestep *t* is independent of what is happening after the timestep t+W. The best solution found for K=36 has 699 color changes, which represents an average of ≈4.6 color changes per radar, or a color change every 39 s.

### 6.3. Example Data Extraction from Optimum

One piece of interesting datum that can be extracted from these solutions concerns how the different colors are distributed among the different radars. For each radar, by looking at the relative position of radars using the same color and displaying it as a distribution (weighted by the distribution of radar itself) we obtain Figure 12. The simulation is on a highway with 2 × 3 lanes, and the different lines seen in the pictures correspond to different lanes relative to the vehicle. The image on the left represents the distribution of same color radar in the opposite lanes, whereas the one on the right represents the distribution for radars in the lanes going the same way.

From the difference between the two, radars with identical color are more often in lanes going in the same direction. The dark area surrounding the center point means that two radars close to each other (below ≈150 m) are very rarely using the same color as they are often in indirect line-if-sight. For lanes going the opposite way, the further away the lane is, the higher the density of same color radars. This is particularly noticeable as the first lane to the left does not have a single same-color radar until ≈500 m away.

These results indicate that to avoid interference while not changing radar parameters to often, it is a good idea to share the available radar band in a way that takes into account the orientation of the radars (or the traffic). Mitigation strategies splitting the available band into two distinct parts depending on the radar orientation have already been simulated in [15,16] and yielded good interference mitigation, but the optimal radar band sharing method seems to require a less strict separation to allow two radars facing in the opposite direction to still use the same waveform, as shown by the non-zero distribution of the picture on the left.

## 7. Conclusions

In this paper, the problem of mitigating automotive radar interference has been translated into a dynamic graph K-coloring problem to find the optimal way to share the available radar band in a given scenario. The metaheuristics proposed in this paper are not implementable in cars in real time as they have a high computational load and require knowledge of the environment that is much too precise. However, they allow for the extraction of optimal radar band sharing to avoid interference. The results yielded are promising for the stability of the system as when applied to our simulation scenario; it avoided all interference while having only 6 changes of parameters out of the 151 radars present during the 3-min simulation. The results were achieved using LRR-FMCW radars, but the metaheuristic itself can be applied to any kind of radar (MRR, SRR, Lidars, SAR, etc.) as long as the principles of line-of-sight and orthogonal resources apply the radar in question.

Two frameworks have been introduced for optimal dynamic K-coloring, on which has been applied two metaheuristics, one with simulated annealing and the other with a genetic algorithm. These metaheuristics allowed one to find valid dynamic K-coloring while minimizing the number of color changes necessary to keep the K-coloring valid. These K-colorings, when applied to the road traffic simulation (2 km of highway, 151 radars), allowed one to avoid all interferences while requiring only 6 waveform changes (with 36 available orthogonal waveforms) during the whole 3 minutes simulation. Decreasing the number of available orthogonal waveforms causes an increase in the number of waveform changes required to avoid all interferences, but it remains low, as until 22 available waveforms, radars need on average only one waveform change during the whole simulation.

The simulated annealing metaheuristic has been used on the non-windowed framework to find a near optimal sharing of the radar band. The resulting K-coloring can offer insight into how to share the available band. The same-color radar distribution shown in Section 6 suggests that the best band sharing involves splitting the band depending on the radar orientation.

The genetic algorithm metaheuristic has been used on the windowed framework to simulate a finite knowledge of the future. The results yielded have more color changes, but they are more achievable in real scenarios.

The metaheuristics presented in this paper open the door to new AI techniques as they can generate optimal radar band sharing that can be used as training data. Future work may involve using a mix of GNN and LSTM to extract the spatial and temporal patterns given by the metaheuristic and make the process applicable in real time. These AI techniques would also be enabled by the progress made in collective perception [38], sensor fusion, and tracking [39] as they allow one to improve the knowledge of the environment used in the metaheuristic to extract the best possible radar band sharing. The uncertainties regarding the environment could be handled by the metaheuristic by replacing the adjacency matrix by a weighted one, where values are between 0 and 1 depending on the probability of line-of-sight.

## Figures and Tables

**Figure 1 sensors-23-05765-f001:**
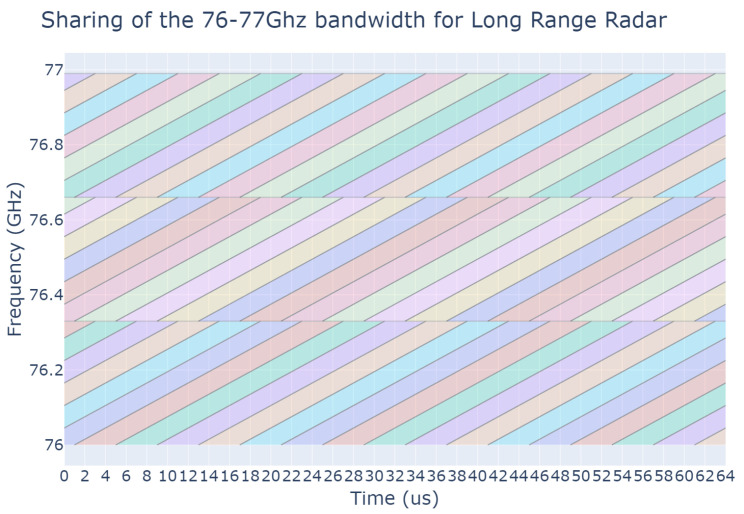
Example of band splitting: the bandwidth is split into 36 different time-frequency zones corresponding to the different colors. They are designed to accommodate an FMCW chirp with a bandwidth of 300 MHz, and a chirp time of 20 μs with a 4 μs reset time and with a receiver bandwidth of 30 MHz (up to 40 MHz). Time/frequency zones are spaced apart by 4 μs in time and 333 MHz in frequency, to fill the 1 GHz available bandwidth. With a duty cycle of 50%, the bandwidth is split into 6 × 3 × 2 = 36 zones.

**Figure 2 sensors-23-05765-f002:**
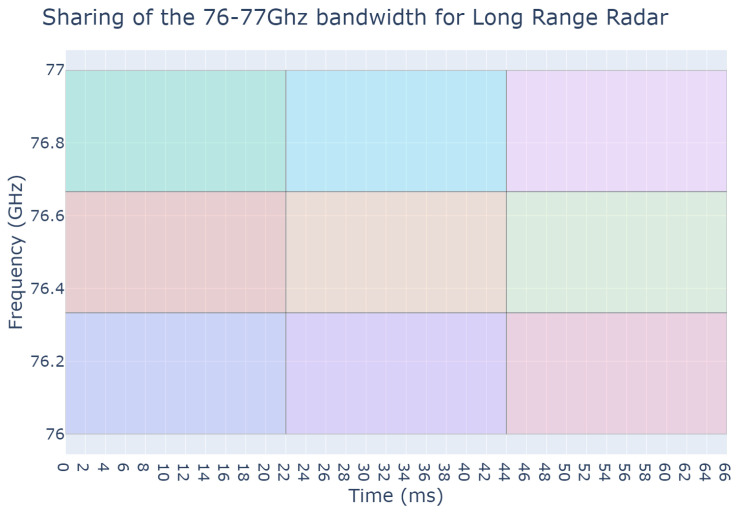
Example of band splitting: the bandwidth is split into 9 different time-frequency zones corresponding to the different colors. They are designed to accommodate a FMCW frame with a bandwidth of around 300 MHz, an emitting time of 22 msm and a duty-cycle of 33% to accommodate 2 other radars in the same frequency band without overlapping emissions. This splits the radar band into 3 × 3 = 9 zones.

**Figure 3 sensors-23-05765-f003:**
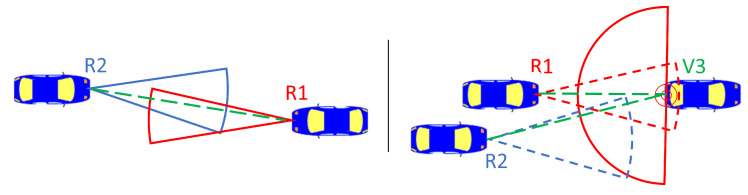
Illustration of line-of-sight (left) where two radars R1 and R2 see each other directly. Illustration of indirect line-of-sight (right) where the signal from radar R1 bounces on the back of vehicle V3, generating an echo at 180° that is in line-of-sight of radar R2.

**Figure 4 sensors-23-05765-f004:**
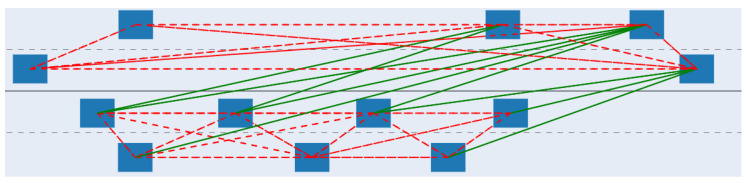
Snapshot from the simulation. Each line-of-sight, direct, (solid green) or indirect (dashed red) between two radars of two cars (blue) are represented with a line.

**Figure 5 sensors-23-05765-f005:**
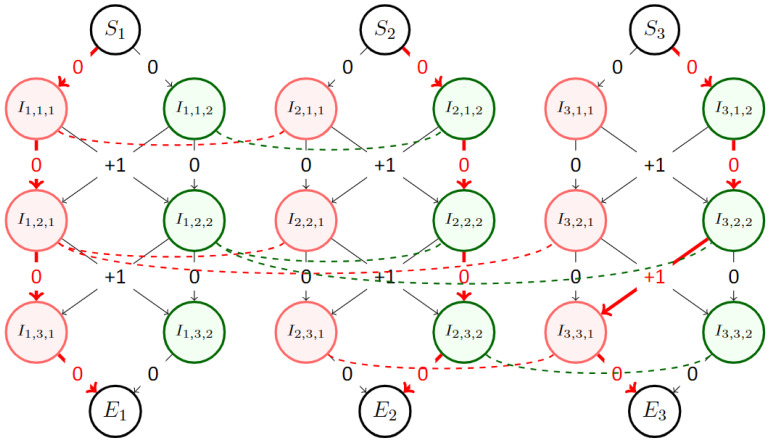
Example of coloring graph with 3 radars, 2 colors, and 3 timesteps, with the following adjacency matrices (denoted with the dotted lines): $A_{0}=\left[\left[0,1,0\right],\left[1,0,0\right],\left[0,0,0\right]\right]$
$A_{1}=\left[\left[0,1,1\right],\left[1,0,0\right],\left[1,0,0\right]\right]$
$A_{2}=\left[\left[0,0,0\right],\left[0,0,1\right],\left[0,1,0\right]\right]$. By using the order [$R_{1}$,$R_{2}$,$R_{3}$] at the start, the shortest path for $R_{1}$ is choosing color 1 and keeping it as there are no conflicts. The shortest path for $R_{2}$ is then to pick color 2 to avoid conflicting with $R_{1}$ at timesteps 1 and 2. Finally, the shortest path for $R_{3}$ is to chose color 2 until timestep 2 to avoid conflicts with $R_{1}$, then change to color 1 to avoid conflicting with $R_{2}$. These paths are denoted by the thick red arrows. In total, this solution yields 0 conflict and 1 color change.

**Figure 6 sensors-23-05765-f006:**
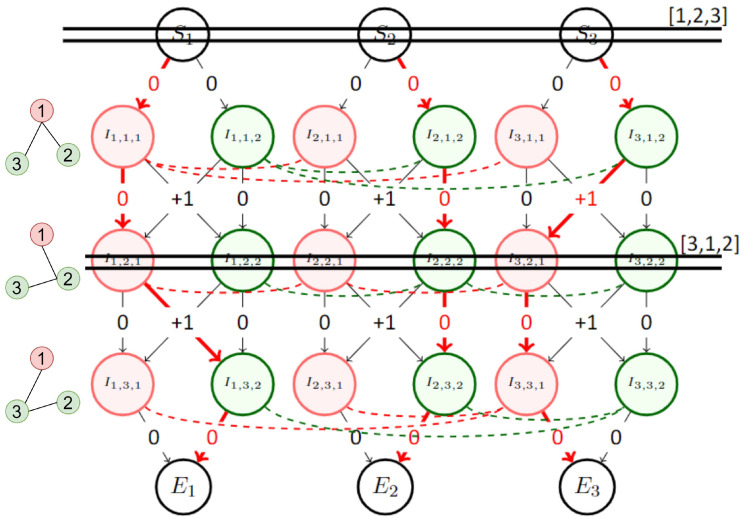
Small example graph to illustrate the necessity of gates. Without adding the second gate, the first sub-graph to compute its shortest path picks the red color and keeps it, and the second sub-graph picks the green color to avoid conflict with the 1st one. The third will have a conflict when it’s adjacent to the two other. By adding a gate in position 2, it is possible to reshuffle the order of shortest path computation, forcing another sub-graph to change its color. In this example, the first gate has the order [1,2,3] and the second one has the order [3,1,2]. The temporal adjacency matrix is illustrated by the small graphs, left of the coloring graph.

**Figure 7 sensors-23-05765-f007:**
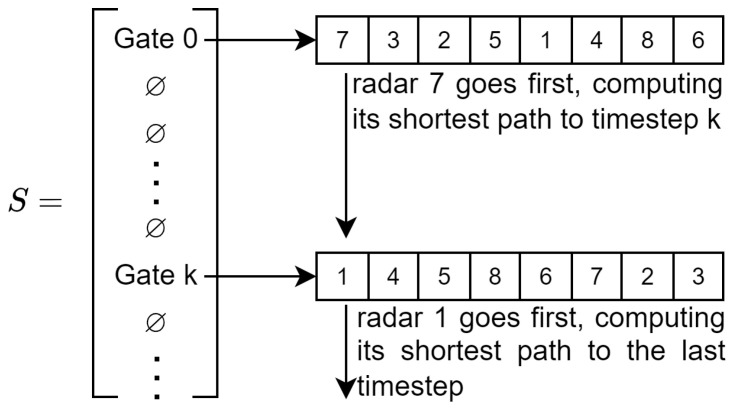
Solutions are in the form of a list of gates. In this example, the list is empty except for timesteps 0 and k, which contain gates with different permutations.

**Figure 8 sensors-23-05765-f008:**
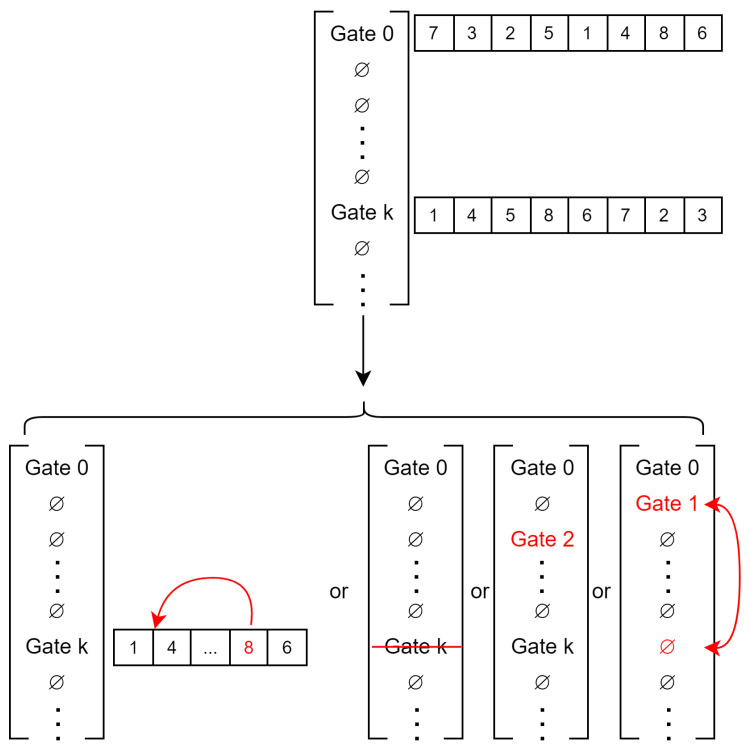
A neighbor of a solution is found by randomly applying one of four different actions. From left to right: changing the order of a gate by selecting a radar and moving it earlier in the order, deleting a gate, adding a random gate, and swapping a gate to another timestep.

**Figure 9 sensors-23-05765-f009:**
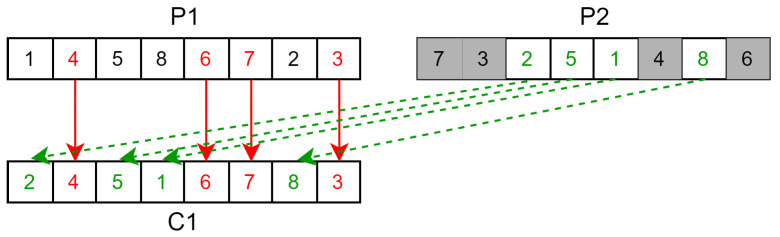
Example of position-based crossover operator (POS). C1 is built by copying a random set from P1 (red) into C1 and filling the blanks following the order of P2 (green).

**Figure 10 sensors-23-05765-f010:**
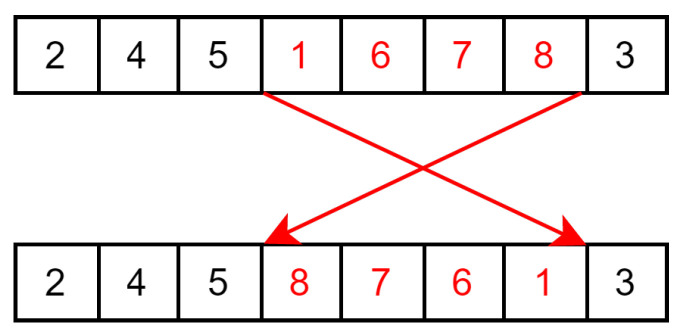
Example of reverse sequence mutation (RSM). A random section of the order is reversed.

**Figure 11 sensors-23-05765-f011:**
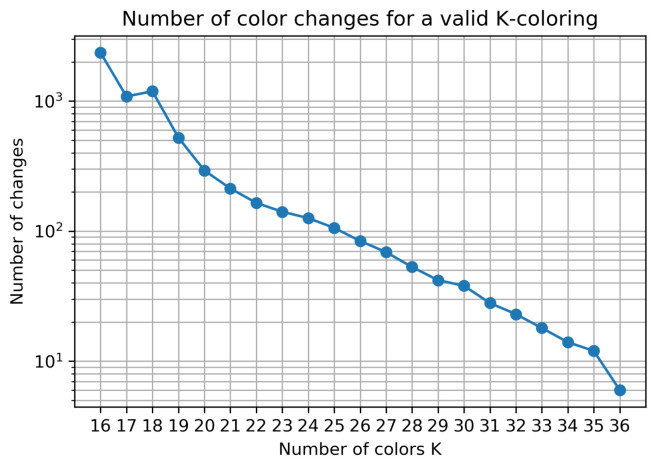
Number of color changes (log scale) in the best solution for different values of *K* (does not include solutions with conflicts).

**Figure 12 sensors-23-05765-f012:**
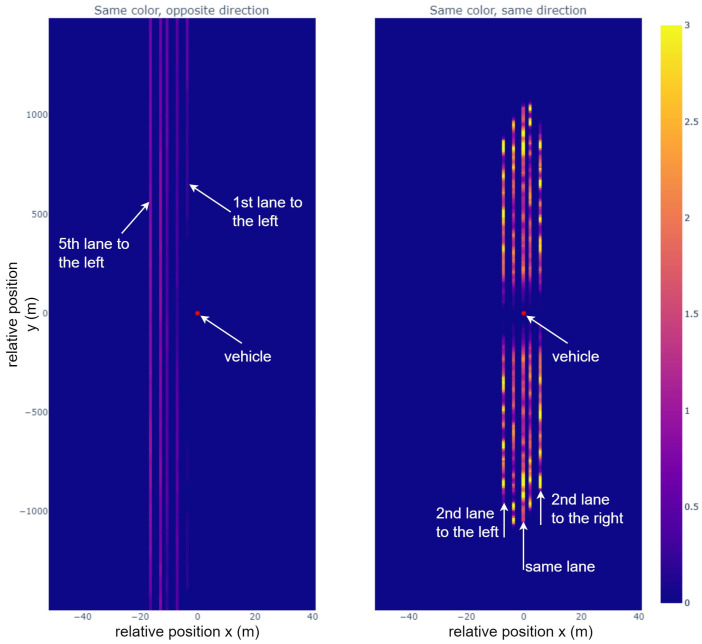
Distribution of radars using the same resource in the best solution found, depending on their relative position and orientation. This distribution is found by dividing, at each position, the average number of radars using the same color by the average number of radars at this position and multiplying this value by the number of available colors. The resulting value at each position is the ratio of radars with the same color over the expect number of radars with the same color (when colors are random). For example, with the color scale in this figure, an area in yellow is an area where radars are three times more likely to have the same color compared to a random color assignment. Radars use the same resources more often when they are oriented in the same direction (right picture) than in opposite one (left picture). Sharing resources with radars in the opposite direction still happens when they are either far away (>500 m) or multiple lanes to the side. An area of ≈150 m (forward and backward) around a radar will contain very few radars using the same resource.

**Table 1 sensors-23-05765-t001:** Parameters used for the simulated annealing version of the metaheuristic.

Parameter	Value
α	0.993
$cost_{edge}$	1
$cost_{conf}^{end}$	2∗T
$cost_{conf}^{init}$	0.001
$n_{iter}$	1000
$T_{end}$	$0.001*T_{start}$

**Table 2 sensors-23-05765-t002:** Results from the simulated annealing with the parameters described in Section 6 for different numbers of available colors. The amount of recoloration in the solutions found by the metaheuristic increases exponentially when lowering the number of available colors *K*. Solutions found have no conflict except with K=16 and K=17.

	Metaheuristic Results		Benchmark Method	
*K*	Conflicts	Changes	Conflicts	Changes
36	0	6	179	159
35	0	12	190	166
34	0	14	203	177
33	0	18	214	184
32	0	23	226	196
31	0	28	245	208
30	0	38	263	219
29	0	42	288	241
28	0	53	307	252
27	0	69	333	266
26	0	84	368	292
25	0	106	405	317
24	0	126	449	352
23	0	141	491	373
22	0	165	554	416
21	0	213	655	477
20	0	293	767	536
19	0	524	916	610
18	0	1190	1148	707
17	10	1086	1709	898
16	22	2357	2664	1165

**Table 3 sensors-23-05765-t003:** Results from the genetic algorithm on the sliding window version with the parameters described in Section 6 for different numbers of available colors. The solutions found have more re-colorations and more conflicts than the non-windowed. The amount of recoloration in the solutions found by the metaheuristic increases exponentially when lowering the number of available colors *K*. Most solutions found have no conflict until K≤18.

*K*	Nb of Conflicts	Nb of Changes
36	0	699
34	0	684
32	0	770
30	0	845
28	0	811
26	0	944
24	0	1123
22	0	1060
20	0	1287
18	123	1403
16	1773	1679

**Table 4 sensors-23-05765-t004:** Parameters used for the genetic algorithm version of the metaheuristic.

Parameter	Value
*W*	10
$Pop_{size}$	100
$p_{mutation}$	0.1
$cost_{edge}$	1
$cost_{conf}^{end}$	2∗W
$cost_{conf}^{init}$	0.001
$n_{iter}$	1000

## Data Availability

Publicly available datasets were analyzed in this study. This data can be found here: https://github.com/Ramboun/dynamic_k_coloring_data (accessed on 1 June 2023).

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
