# Peer review of "Metaheuristic for Optimal Dynamic K-Coloring Application on Band Sharing for Automotive Radars"

_sensors, 2023, doi:10.3390/s23125765_

Round 1
Reviewer 1 Report
The authors are describing a heuristic method for dynamic graph coloring with intended application to reducing interference in automotive radars. The paper is well-written, and even though highly mathematical, it is understandable by a generally knowledgeable person in the field.
My largest concern is about the simulation environment which involves SUMO and which is not well described. I understand that the authors lean on the previous publications [12] and [27], but more details would nevertheless be necessary. In my opinion this part should be strengthened. I praise including the actual graph which was extracted from the traffic simulator on https://github.com/Ramboun/dynamic_k_coloring_data page but it would be much more beneficial to include the SUMO scripts so that the readers could reproduce the graph from the traffic simulation, then change the simulation, generate new temporal adjacency matrices and run the proposed coloring algorithms. Implementation of algorithms could also be given there in the actual code instead of just pseudo code in the appendix.
Minor notes and suggestions include:
- in figures 1 and 2 titles do not abbreviate Long-Range Radar as LRR and write y-axis in GHz instead of e+10.
- in Figure 4. better use dashed or dotted (red) instead of solid (red) lines for indirect line-of-sight (ILOS)
- name the Bellman algorithm as Bellman-Ford algorithm, as it is more common
English mistakes or typos include:
line 24: "a rapid increase of radar on the road" instead "a rapid increase on radar on the road"
line 57: "by each radar" instead of "by each radars"
line 114: "the total number of colors used" instead of "the amount of color used"
always use: "an FMCW" instead of "a FMCW"
line 184: "this paper views radars" instead of "this paper view radars"
line 473: "line-of-sight" instead of "line-if-sight"
line 476: "These results indicate" instead of "These results indicates"
line 481: "sharing methods seem to" instead of "sharing methods seems to"
Reviewer 2 Report
|
Journal |
Sensors |
|
Title |
Metaheuristic for Optimal Dynamic K-Coloring, application on band sharing for Automotive Radars |
|
Authors |
Sylvain Roudiere, Vincent Martinez, Pierre Maréchal and Daniel Delahaye |
|
Comments |
This paper proposed a method named metaheuristic for optimal dynamic k-coloring for automotive radars. It is interesting, but some issues must be resolved for possible publication and to be reviewed again. |
|
1. In the Abstract, please do not cite papers, like “as shown in our previous work [1]”. 2. In the introduction, this reviewer is interested whether synthetic aperture radar can be used in automotive fields and should be added and considered in the end of this paper, such as mask attention interaction and scale enhancement network. Please find and consider this type used for SAR ship applications. 3. About 88 GHz radars, the road-side radars should be reviewed, which are key steps for automotive driving, and what is the difference between 3d super-resolution imaging method for distributed millimeter-wave automotive radar system. The MW-Radar can be improved using synthetic aperture and MIMO. 4. Vehicle to everything (V2X) is popular now. The wireless communication should be combined with it, such as 5.5G and 6G. Moreover, the authors should think whether ship traffic can become ship to everything (S2X), where a lot of works have been made in this field, for example, some data like sar ship detection dataset (ssdd) and ls-ssdd-v1.0, and some early works like high-speed ship detection in sar images based on a grid convolutional neural network and depthwise separable convolution neural network for high-speed sar ship detection. These should be treated seriously in this paper end. 5. For fig.2, why do the authors select LRR instead of MRR? 6. For fig. 5, why do not the authors use graph network in the deep learning community and the lstm used in polarization fusion network with geometric feature embedding for sar ship classification. The authors should compare them or state them anywhere. 7. Fig. 6 is similar the fpn used in object detection, some classical work like quad-fpn quad feature pyramid network. Please clarify them in the end of this paper. 8. IMHO, the Conclusion should be re-written to 1) explicitly describe the essential features/advantages of the review that other reviews do not have, and 2) describe the limitation(s) of the review. The English should be improved greatly. 9. The English should be improved. |
Journal Sensors
Title Metaheuristic for Optimal Dynamic K-Coloring, application on band sharing for Automotive Radars
Authors Sylvain Roudiere, Vincent Martinez, Pierre Maréchal and Daniel Delahaye
Comments This paper proposed a method named metaheuristic for optimal dynamic k-coloring for automotive radars. It is interesting, but some issues must be resolved for possible publication and to be reviewed again.
1. In the Abstract, please do not cite papers, like “as shown in our previous work [1]”.
2. In the introduction, this reviewer is interested whether synthetic aperture radar can be used in automotive fields and should be added and considered in the end of this paper, such as mask attention interaction and scale enhancement network. Please find and consider this type used for SAR ship applications.
3. About 88 GHz radars, the road-side radars should be reviewed, which are key steps for automotive driving, and what is the difference between 3d super-resolution imaging method for distributed millimeter-wave automotive radar system. The MW-Radar can be improved using synthetic aperture and MIMO.
4. Vehicle to everything (V2X) is popular now. The wireless communication should be combined with it, such as 5.5G and 6G. Moreover, the authors should think whether ship traffic can become ship to everything (S2X), where a lot of works have been made in this field, for example, some data like sar ship detection dataset (ssdd) and ls-ssdd-v1.0, and some early works like high-speed ship detection in sar images based on a grid convolutional neural network and depthwise separable convolution neural network for high-speed sar ship detection. These should be treated seriously in this paper end.
5. For fig.2, why do the authors select LRR instead of MRR?
6. For fig. 5, why do not the authors use graph network in the deep learning community and the lstm used in polarization fusion network with geometric feature embedding for sar ship classification. The authors should compare them or state them anywhere.
7. Fig. 6 is similar the fpn used in object detection, some classical work like quad-fpn quad feature pyramid network. Please clarify them in the end of this paper.
8. IMHO, the Conclusion should be re-written to 1) explicitly describe the essential features/advantages of the review that other reviews do not have, and 2) describe the limitation(s) of the review. The English should be improved greatly.
9. The English should be improved.
Reviewer 3 Report
Despite radar specifications set forth by standardizing organizations like ETSI, which establish a maximum limit for transmission power, these standards don't extend to specific radar waveform parameters or channel access scheme policies. This lack of mandatory guidelines potentially increases the risk of interference as radar adoption continues to grow. In this work, a metaheuristic is proposed to find the optimal resource sharing between radars, knowing their relative positions and thereby the line-of-sight and non-line-of-sight interference risks during a realistic scenario. This method can be very useful for simulated scenarios to find a near-optimal sharing of the radar band that can then be used to extract patterns or as data generation for machine learning. The topic in this paper is pretty interesting for the readers and the contributions are sufficient to be published in this journal. I only have some small comments to improve the paper.
A. Please highlight the contributions in a comparable manner in the introduction. In the current form, authors only stated what works have been done in this paper rather highlight the novelty or merits of the method in the paper;
B. In the simulation environment section, it was mentioned the sumo software is used to generate the traffic flow, which is good to me. In the most recent literature, works have been done to collect more realistic data based on the real connected automated vehicle such as in: an automated driving systems data acquisition and analytics platform. Please discuss this work in the paper as well to attract more interest from the connected automated vehicle community.
C. Authors mentioned that the centralized approach assumes that everything about the system is known (e.g., past and future positions of the vehicles). However, in real applications, this information is hard to obtain. They are based on localization and object detection and tracking algorithms which may introduce uncertainties to the vehicle position or historical trajectory. Please consider including some works: hydro-3d: hybrid object detection and tracking for cooperative perception using 3D lidar; autonomous vehicle kinematics and dynamics synthesis for sideslip angle estimation based on consensus kalman filter, in regard to discussing the potential of how to incorporate the uncertainty to design the algorithm in this paper.
D. It will be great if the authors can discuss more about the future work of this paper. In addition, from my understanding, the work in this paper can be generalized into the lidar-based object detection community such as in: an automated driving systems data acquisition and analytics platform, as the lidar also encounters the interference issue. I hope the authors can mention this in the paper.
Round 2
Reviewer 1 Report
The authors have addressed all the concerns I had, hence I recommend the paper to be published.
Reviewer 2 Report
Accept in present form. No more comments.
Fine.